# Mechanistic basis of atypical *TERT* promoter mutations

Kerryn Elliott [1], Vinod Kumar Singh [1], Alan Bäckerholm [1], Linnea Ögren [1], Markus Lindberg [1], Katarzyna M. Soczek[2], Emily Hoberg[1], Tom Luijts [1,3,4], Jimmy Van den Eynden[3,4], Maria Falkenberg[1], Jennifer Doudna [2], Anders Ståhlberg [5,6,7] & Erik Larsson [1] ✉

Non-coding mutations in the *TERT* promoter (*TERT*p), typically at one of two bases −124 and −146 bp upstream of the start codon, are among the most prevalent driver mutations in human cancer. Several additional recurrent *TERT*p mutations have been reported but their functions and origins remain largely unexplained. Here, we show that atypical *TERT*p mutations arise secondary to canonical *TERT*p mutations in a two-step process. Canonical *TERT*p mutations create de novo binding sites for ETS family transcription factors that induce favourable conditions for DNA damage formation by UV light, thus creating a hotspot effect but only after a first mutational hit. In agreement, atypical *TERT*p mutations co-occur with canonical driver mutations in large cancer cohorts and arise subclonally specifically on the *TERT*p driver mutant chromosome homolog of melanoma cells treated with UV light in vitro. Our study gives an in-depth view of *TERT*p mutations in cancer and provides a mechanistic explanation for atypical *TERT*p mutations.

Reactivation of telomerase is a key mechanism that allows cancer cells to divide continuously without undergoing senescence or cell death[1]. In 2013, two studies demonstrated that point mutations in the Telomerase reverse transcriptase (*TERT*) promoter (*TERT*p) can activate *TERT* transcription, thereby increasing levels of the gene product, the catalytic subunit of telomerase[2,3]. *TERT*p mutations are now known to be among the most frequent somatic driver events in human cancer and have been reported in > 50 cancer types[4–7].

Recurrent *TERT*p mutations normally occur as mutually exclusive C > T single nucleotide variants (SNVs) at one of two specific positions, −124 and −146 bp upstream of the ATG start codon (also known as C228T and C250T, respectively). These mutations both create a de novo activating binding site for E26 transformation specific (ETS) family transcription factors (TFs), and both result in

formation of an identical sequence, 5′-CCCC**TTCCG**G-3′ (ETS motif in bold). Mechanistically, the ETS protein GABPA has been proposed to bind the de novo sites as a heterotetramer together with GABPB while interacting with a native (pre-existing) ETS site in the *TERT*p (known as ETS-200), positioned such that the helical phase is preserved[8,9]. The −124 and −146 bp regions are thus "proto-ETS sites" primed to be mutationally converted into activating functional sites that recruit the GABP/ETS complex, in both cases through interaction with a native ETS site.

While recurrent *TERT*p mutations are found exclusively at the −124 and −146 bp positions in most cancer types, increasingly larger whole genome and targeted panel sequencing studies have also uncovered several atypical recurrent *TERT*p mutations, often in melanoma or other skin cancers (Supplementary Table 1). The most

[1]Department of Medical Biochemistry and Cell Biology, Institute of Biomedicine, The Sahlgrenska Academy, University of Gothenburg, Gothenburg, Sweden. [2]Department of Molecular and Cell Biology, University of California, Berkeley, CA, USA. [3]Department of Human Structure and Repair, Ghent University, Ghent, Belgium. [4]Cancer Research Institute Ghent, Ghent, Belgium. [5]Department of Clinical Genetics and Genomics, Sahlgrenska University Hospital, Region Västra Götaland, Gothenburg, Sweden. [6]Wallenberg Centre for Molecular and Translational Medicine, University of Gothenburg, Gothenburg, Sweden. [7]Sahlgrenska Center for Cancer Research, Department of Laboratory Medicine, Institute of Biomedicine, The Sahlgrenska Academy, University of Gothenburg, Gothenburg, Sweden. ✉e-mail: erik.larsson@gu.se

frequent ones include non-ETS-forming C > T SNVs at −126 and −149 bp, close to the canonical driver mutations, and at positions flanking the native ETS site[2,10–16]. Additionally, a rare recurrent CC > TT dinucleotide variant (DNV) at −139/−138 bp introduces a de novo ETS site shown to activate *TERT* transcription in reporter assays[17]. Despite repeated reports of recurrent mutations at several additional positions, the role of such non-canonical *TERT*p mutations and the reason for their preferential emergence in skin cancers has remained elusive.

Studies dating back to the 1980's, as well as recent investigations using sequencing-based methods, have shown that many TFs have the ability, through their impact on DNA geometry and flexibility, to modulate DNA damage formation by UV light[18–22]. Of note, ETS factors have been shown to induce particularly favourable conditions for formation of cyclobutane pyrimidine dimers (CPDs), the principal UV DNA damage lesions, leading to strong C > T mutation hotspots at numerous ETS binding sites (TTCCK motifs, where K is G/T) in skin cancers[23–27]. These mutations form primarily at the two bases preceding the core motif or, to a lesser extent, at the first central cytosine[24,27]. Mutations have been shown to form at highly elevated frequency at such sites following UV irradiation in vitro, further supporting that ETS hotspot mutations are generally passengers rather than due to positive selection[23,25]. Structural analysis of ETS-bound DNA demonstrated DNA conformational changes that facilitate CPD formation[24].

Here, we use available *TERT*p targeted sequencing data, melanoma whole genome sequencing data, and in vitro experiments to explain the aetiology of non-canonical *TERT*p mutations. Our analyses support that these typically arise in a two-step process, wherein a canonical *TERT*p driver mutation at a proto-ETS site, allowing ETS to bind, represents a first mutational hit. This sets the stage for a second hit in the form of a passenger UV hotspot mutation at either the newly formed ETS site or the native ETS site, owing to the ability of ETS factors to stimulate UV damage formation where they bind. Our study gives an overview of *TERT*p mutations in human cancers and clarifies the origin of most atypical recurrent *TERT*p mutations.

## Results

### Most atypical *TERT* promoter mutations are restricted to skin cancers

To obtain an overview, we probed 19,755 independent tumours from a broad range of *TERT*p-mutated (≥1% of samples) human cancers with the help of targeted sequencing data in GENIE[28] (Supplementary Table 2). Recurrent *TERT*p mutations were observed predominantly at the −124 and −146 bp positions as expected, and were most prevalent in cancers arising from cells with low rates of self-renewal such as gliomas (reaching up to 96% in anaplastic oligodendroglioma) as well as in skin cancers (77% of skin cutaneous melanomas, SKCM), as previously reported (Fig. 1)[29]. *TERT*p mutation frequency varied considerably, with cancer types such as lung and colon adenocarcinoma being at the other end of the spectrum (1.1% and 1.8%, respectively), also in agreement with prior findings[30]. While the −124 bp mutation dominated over −146 bp in most *TERT*p mutated cancer types, skin cancers exhibited the two mutations at almost equal frequency (Fig. 1).

Additional recurrent *TERT*p mutations (present in ≥10/19,755 samples) were essentially restricted to skin cancers exhibiting a high fraction of UV light-derived mutations (single and double base mutational signatures SBS7 and DBS1, respectively). These included the ETS-forming DNV at −139/−138 bp, the SNVs at −149 and −126 bp, and at −101 and −100 bp overlapping the native ETS site (Fig. 1). The recurrent mutations were almost exclusively (99.1%) C > T SNVs or CC > TT DNVs in dipyrimidine sequence contexts, compatible with UV mutagenesis. Together, these results implicate UV-related mutational processes as a source of atypical *TERT*p mutations.

### Atypical *TERT*p events arise at UV damage hotspot positions within the ETS motif

Given the presence of proto-ETS sites (convertible to functional sites) as well as a native (preexisting) ETS site in the *TERT*p, we hypothesized that elevated UV damage and increased mutation rate at ETS binding sites may play a role in the aetiology of atypical *TERT*p mutations[24–27]. To test this, we defined a subcohort, "High *TERT*p, UV" (1,569 samples), comprising of cancer types having a high average *TERT*p mutation frequency and high UV mutation burden, while still exhibiting considerable inter-sample variability in these variables (Fig. 1). These were contrasted against a "High *TERT*p, no UV" subcohort (4,065 samples), with similar levels of *TERT*p mutations but lacking UV exposure (Fig. 1).

The recurrent *TERT*p mutations were limited to a small, 56 bp, upstream region, and nearly all atypical mutations were found in the UV-exposed set as expected (Fig. 2a, Supplementary Table 3). Of these, the most common were two ETS-forming CC > TT DNVs: −139/−138 bp, previously suggested to be a driver event[2,17], and −125/−124 bp, a driver at the canonical −124 bp (C228T) site[2,3] (Fig. 2a). Presumably, strongly elevated CC > TT DNV frequency in UV-exposed cells provides a substrate for positive selection acting on DNVs at these sites in skin cancers (Fig. 1, Supplementary Table 4)[31].

Of the remaining, non-ETS-forming, atypical mutations, the most frequent were at the native ETS site, at cytosines upstream of the ETS binding motif known to exhibit a strong UV damage hotspot effect when occupied[7,23–25,27] (Fig. 2a, indicated in the sequence). Specifically, SNVs or DNVs at −101/−100 bp (CCTTCCG; mutations underscored, ETS motif bolded) or to a lesser extent at −97 bp (CCTTCCT), a position in the motif known to exhibit a weaker UV hotspot effect[25], were detected in 44/1,569 samples (2.8%) in the UV-exposed subcohort but were absent in the non-exposed set. Similarly, atypical mutations at −149 bp and, more rarely, at −148 bp were found in 22/1,569 samples (1.4%) in the UV-exposed set while being absent in non-exposed samples (Fig. 2a). These positions are just upstream of the −146 bp (C250T) canonical proto-ETS driver site compatible with a CPD hotspot effect at this locus. In the same way, variants at −126 bp, present in 14/1,569 samples (0.9%), arise at a predicted hotspot position upstream of the −124 bp (C228T) canonical proto-ETS site (Fig. 2a). The observed patterns of mutagenesis, as well as the association with UV-exposure, thus suggest that the majority of atypical SNVs may be passengers arising due to elevated UV damage at *TERT*p ETS sites.

### Mutation co-occurrence implies a two-step progression

The canonical ETS-forming driver mutations at −146 and −124 bp activate *TERT* by recruitment of a larger ETS dimer or multimer complex that interacts both with the de novo-formed and the native ETS site (proposedly GABPA/GABPB or ETS1/p52)[8,9,32] (illustrated in Fig. 2a). As ETS sites should only exhibit elevated UV mutagenesis when occupied, we next subdivided the UV-associated subcohort by presence or absence of a primary *TERT*p driver mutation (Fig. 2b).

In the UV-exposed set, all 22 samples with mutations at −149 or −148 bp also carried a corresponding (−146 bp) primary driver event ($P = 4.5 \times 10^{-12}$; Fig. 2b, Supplementary Fig. 1). Of the 14 mutations at −126 bp, 10 occurred in samples having the corresponding (−124 bp) driver event ($P = 9.4 \times 10^{-4}$), while the remaining 4 co-occurred with −146 bp events (Fig. 2b, Supplementary Fig. 1). The closely spaced co-occurring mutations were called together as oligonucleotide variants (ONVs), thus linking them to the same chromosome homologue, in some but not all cases, potentially explained by heterogenous ONV calling in the GENIE cohort[33] (Fig. 2b, Supplementary Table 3). Furthermore, nearly all (41/44; 93.2%) native site mutations occurred in samples also having primary driver events, present in 67.4% of all samples in this subcohort, in agreement with the native site only being occupied in tumours following a *TERT*p driver mutation ($P = 6.0 \times 10^{-5}$; Fig. 2b, Supplementary Fig. 1). Together, these results support that

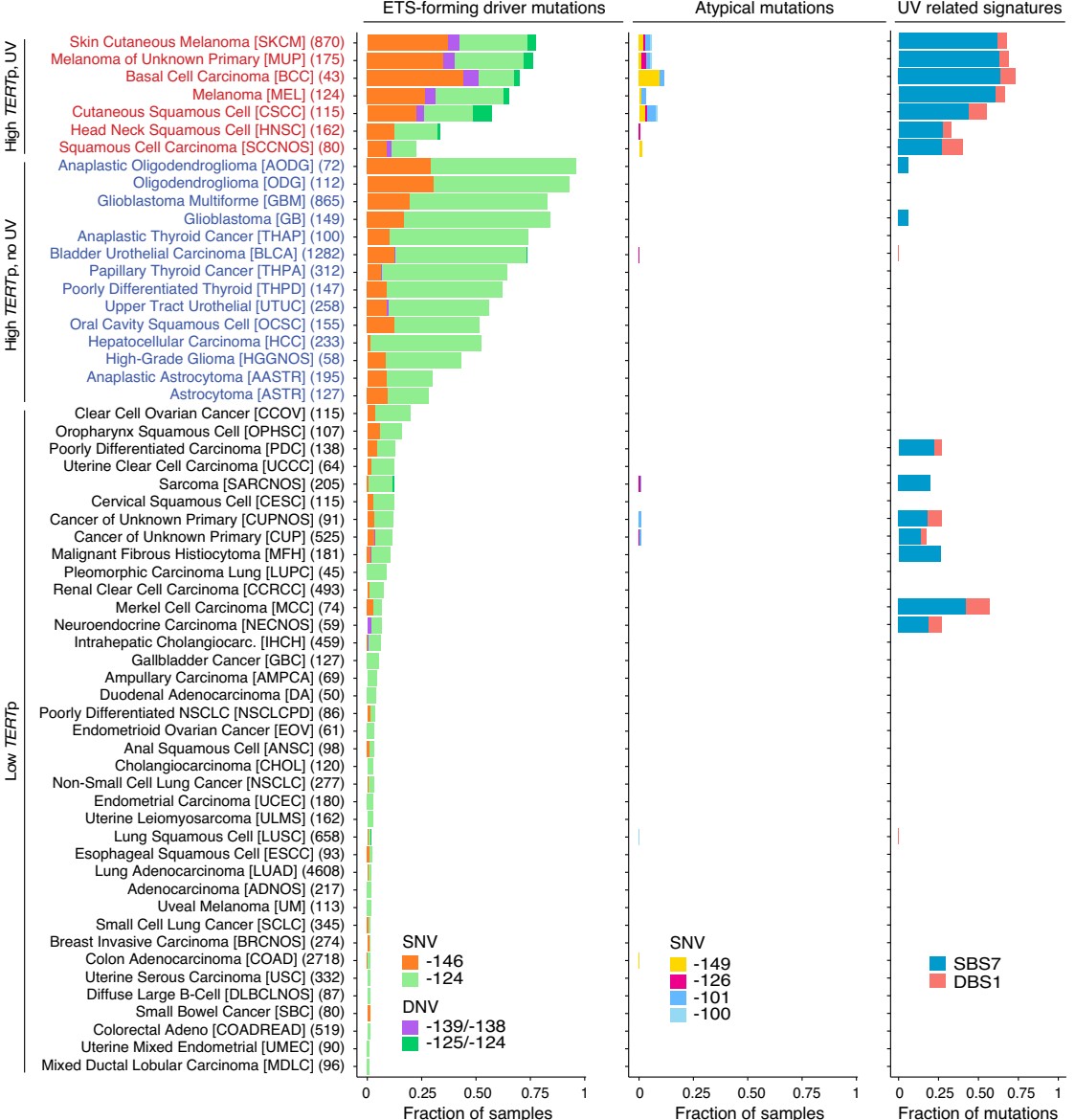

**Fig. 1 | Overview of cancer types with *TERT*p mutations in GENIE.** 59 cancer types from GENIE (v11) were considered, all having at least 1% *TERT*p-mutated samples. For each cancer type, the fraction of samples with ETS-forming *TERT*p driver mutations and recurrent atypical mutations is indicated, as well as the fraction UV-related single and double base substitutions (mutational signatures SBS7 and DBS1, respectively). All recurrent *TERT*p mutations (present in ≥10/19,755 samples and being within 300 bp upstream of the *TERT* translation start site) are indicated by separate colours. Numbers (e.g. −146) refer to base positions relative to the translation start. Cancer types were divided into subcohorts based on the frequency of *TERT*p driver mutations (above or below 20%) and the fraction UV-related substitutions (above or below 10%). The number of patients (each represented by a single tumour sample) is indicated within parentheses for each cancer type, and OncoTree[48] codes are shown within square brackets. SNV, single nucleotide variant; DNV, double nucleotide variant. Source data are provided as a Source Data file.

most atypical *TERT*p SNVs arise in a two-step sequence, wherein a primary driver event first recruits the ETS/GABP complex thus paving the way for a second hit in the form of a UV-induced hotspot mutation at a now-occupied ETS site.

**Mutation distribution supports neutral selection for non-ETS-forming atypical mutations**

Our results suggest that non-ETS-forming atypical *TERT*p mutations are passenger events. To further test this, we binned the *TERT*p and UV-associated skin cancers in GENIE ("High *TERT*p, UV" subcohort in Fig. 1) by UV mutation burden (C > T substitutions in dipyrimidine contexts) and investigated the distribution of *TERT*p mutations across the bins. We found that the frequencies of ETS-forming *TERT*p

mutations, both the canonical drivers and the rare DNVs seen in skin cancers, showed relatively limited correlation with UV mutation burden (r = 0.44, P = 0.45, all combined) and were thus not markedly dictated by the degree of UV mutagenesis, as noted previously for driver mutations in melanoma (Fig. 3a)[23,34]. In contrast, the atypical *TERT*p mutations correlated strongly with UV mutation burden across samples (r = 0.99, P = 1.3 × 10⁻³, all combined), as would be expected for UV-induced mutations under neutral selection (Fig. 3b)[23]. Even within the group of UV-associated skin cancers, atypical mutations thus occur preferably in highly UV-mutated samples. These results are consistent with ETS-forming *TERT*p mutations being drivers and with non-ETS-forming atypical events arising passively due to UV exposure thus being passengers.

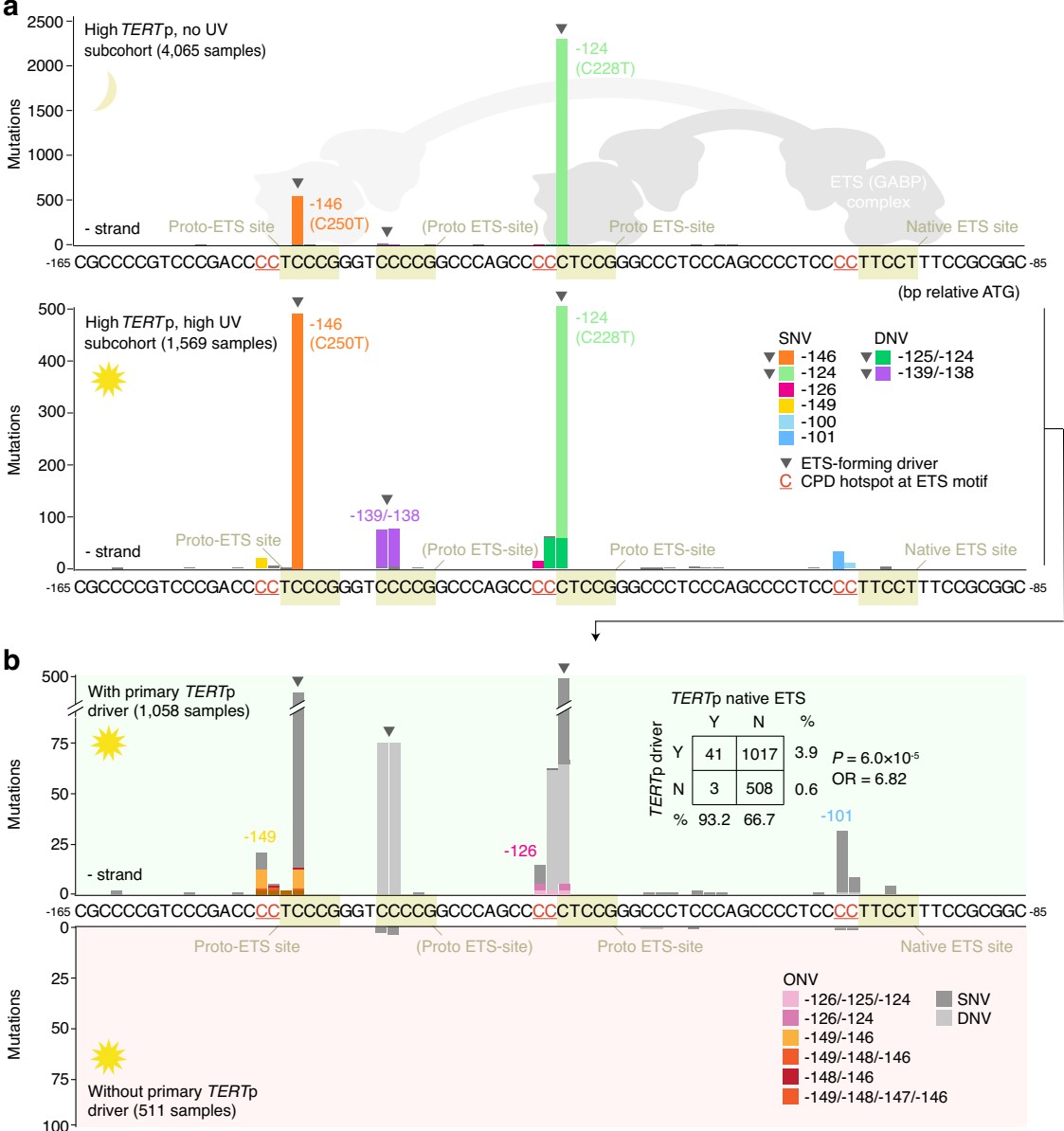

**Fig. 2 | Atypical *TERT*p mutations arise at pre-existing or de novo-formed UV hotspot positions. a** Mutation frequencies in a 80 bp region of the *TERT*p (chr5:1295188–1295268, hg19) in highly *TERT*p-mutated (≥20% of samples) GENIE cancer types, which were further subdivided into non-UV associated ("High *TERT*p, no UV") or UV-associated ("High *TERT*p, UV") subcohorts as indicated in Fig. 1. Mutations at a native (preexisting) ETS factor binding site arise only in the UV-exposed set, at positions in the ETS motif known to exhibit a UV damage hotspot effect when occupied (primarily CCTTCCK but also CCTTCCK). The schematic shows GABP binding to mutant *TERT*p sequences. **b** Frequencies in the UV-exposed subcohort following further subdivision of tumours by presence (positive axis) or absence (negative axis) of a *TERT*p driver mutation (−124, −126 or −139/−138 bp). Nearly all native ETS site mutations co-exist with primary driver events in the same patients, suggestive of a two-step model. Inset shows contingency table for driver and native site mutation co-occurrence (*P*-value from two-sided Fisher's exact test). Equally, atypical mutations at −149 and −126 bp, predicted secondary hotspot bases at de novo ETS sites from −146 and −124 bp driver SNVs, co-exist with driver events. SNV, single nucleotide variant; DNV, double nucleotide variant; ONV, oligonucleotide variant; CPD, cyclobutane pyrimidine dimer. Source data are provided as a Source Data file.

## Allele phasing molecularly links atypical *TERT*p mutations to primary drivers

The proposed two-step model implies that canonical *TERT*p mutations and secondary atypical events should co-exist on the same chromosome homologue. While GENIE provided evidence for mutation co-occurrence at the patient level, allele phase information was limited due to lack of access to raw sequencing data and possible heterogenous protocols for calling of complex variants[33].

To address this, we screened for *TERT*p mutations in 335 melanomas subjected to whole genome sequencing within the Genomics England 100 k Genomes Project. Of nine identified cases with UV hotspot mutations at the native ETS site (six at −101 bp and three at −100 bp), eight also had a canonical primary *TERT*p driver event, in agreement with the results from GENIE (Fig. 4a). Similarly, three cases mutated at the −149/−148 bp secondary UV hotspot site all carried −146 bp primary driver events (Fig. 4a). Interestingly, one of these also carried a −126 bp mutation, again mirroring the GENIE analysis. An additional sample mutated at the −126 bp secondary hotspot site simultaneously carried a −124 bp primary driver event. The atypical events (−101, −100, −149 and −126 bp) were more prevalent in high-UV

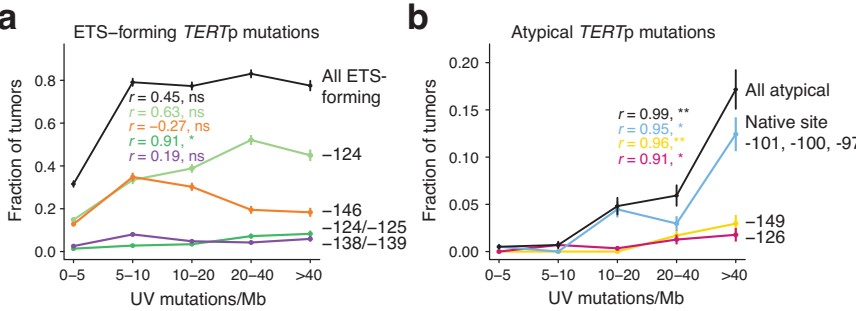

**Fig. 3 | Passenger-like distribution of atypical *TERT*p mutations.** Line plots showing the frequency of key *TERT*p mutations as a function of UV mutation burden (C > T mutations in dipyrimidine contexts) across the *TERT*p mutated, UV-exposed, GENIE subcohort ("High *TERT*p, UV" cancer types in Fig. 1). 1,569 samples were divided into five bins based on burden as indicated on the x-axis (586, 287, 291, 236 and 169 samples per bin). **a** ETS-forming driver mutations show limited correlation with UV mutation burden. **b** The frequency of non-ETS-forming atypical mutations is strongly dictated by UV mutation burden, as expected for UV-induced passengers under neutral selection. Error bars indicate 80% confidence intervals based on the standard errors of the proportions. The Pearson correlation coefficient *r* and associated two-sided *P*-value, calculated based on the average burden in each bin, is indicated for each category. ns, not significant; *P < 0.05; **P < 0.01. Source data including exact *P*-values are provided as a Source Data file.

samples, as expected for UV-induced passengers, while established positively selected driver events were more uniformly distributed, again corroborating our results from GENIE (Fig. 4a).

We next phased the atypical *TERT*p mutations relative to primary driver events in samples where these co-occurred (12 cases described above), by extracting sequencing reads that covered all the key positions (Supplementary Fig. 2). We found that mutations at the native ETS site were always in cis with a primary mutation thus having arisen on the same chromosome homologue (Fig. 4b). Similarly, atypical mutations at −149, −148 bp and −126 always occurred in cis with primary driver mutations (Fig. 4b). Compared to primary mutations, secondary events had similar or only slightly lower allele frequencies, supporting that they occurred early before expansion of major cell clones (Fig. 4b). A sequential model is thus further supported by direct molecular linkage between primary and atypical secondary events in UV-exposed melanomas.

### Elevated UV damage formation at *TERT*p ETS sites upon GABP binding

To test whether ETS binding stimulates UV damage in the *TERT*p, we determined CPD formation levels following UV irradiation of radiolabelled wild-type or mutant *TERT*p dsDNA fragments encompassing the de novo and native sites, with or without recombinant GABP complex. Binding of GABP to the *TERT*p fragments was first established by gel-shift experiments (Supplementary Fig. 3) as well as DNase I footprinting, which revealed interactions at the −124 bp and −146 bp proto-sites specifically when mutated as expected (Fig. 5a, Supplementary Fig. 4). Although the wild-type *TERT*p is considered incapable of significant GABP complex recruitment in cells[8,9], we found that all sequences interacted with recombinant GABP via the native site in this in vitro system (Fig. 5a).

CPD profiling by T4 endonuclease V digestion revealed CPD damage to be consistently elevated at the predicted hotspot CC positions upstream of the mutated, but not wild-type, −146 and −124 bp proto-ETS sites, only in the presence of GABP as predicted (Fig. 5b, Supplementary Fig. 5). Weak yet consistent GABP-dependent CPD induction upstream of the native site was also seen in all fragments, in agreement with the protein binding data (Fig. 5b). Atypical *TERT*p SNVs thus generally arise at base positions exhibiting elevated UV damage formation following GABP binding in vitro.

### Driver-linked secondary mutations arise following UV irradiation in cultured cells

To recapitulate the mutagenic process for atypical *TERT*p mutations in cellular conditions, we first developed an amplicon sequencing assay to enable detection of low-frequency, subclonal, mutations in the *TERT*p. The *TERT*p is notoriously GC rich (>80%), making PCR-based assays challenging[35,36]. Here, we were able to combine SiMSen-seq ultrasensitive amplicon sequencing[37] with a specific polymerase and a GC enhancer additive to sequence the *TERT*p at high fidelity. We next treated A375 melanoma cells, which carry a heterozygous -146 bp C > T (C250T) *TERT*p driver mutation, with daily low doses of UVC light during 6 weeks and compared mutation frequencies in untreated and UV-exposed cells using our SiMSen-seq assay. All reads were then phased in relation to mutation status at the −146 bp position to determine if mutations were arising on the *TERT*p mutant (C250T) or wild-type chromosome homologue (Fig. 6a).

A total of 15,923 and 12,006 error-corrected consensus reads were obtained for these two conditions, respectively (minimum 3 reads oversampling). In non-exposed cells, only a small number of atypical recurrent *TERT*p mutations were detected, present in at most four consensus reads, likely attributable to sequencing noise or pre-existing genetic heterogeneity (Fig. 6b). In contrast, subclonal recurrent mutations were found at expected hotspot positions after UV exposure, at the native ETS site (−101 and −100 bp) and at −149 and −148 bp, which are predicted secondary UV hotspot bases at the −146 bp proto-ETS site (Fig. 6c). Notably, these atypical mutations were almost perfectly phased together with the −146 bp driver event: 21/23 native site mutations and 21/21 mutations at −149/−148 bp occurred in cis with the −146 bp mutation. Atypical *TERT*p mutations frequently seen in UV-exposed cancers can thus be induced by UV exposure in cultured cells, but only when preceded by an ETS-recruiting *TERT*p driver mutation on the same chromosome homologue.

## Discussion

Somatic *TERT*p mutations are the most frequent driver alterations in many cancers and are known to emerge early during oncogenesis, thus being of fundamental importance in human tumour development with several possible uses as prognostic biomarkers[1,38–40]. Our analyses place recurring *TERT*p mutation events into three main categories, the first being the canonical driver C > T SNVs at −146 bp (C250T) and −124 bp (C228T), which activate the *TERT*p through formation of de novo ETS TF factor binding sites. The second encompasses additional, and much less frequent, driver events in the form of CC > TT DNVs at −125/−124 bp and −139/−138 bp, the former being functionally equivalent to the −124 bp driver while the latter is ETS-forming at an additional, third, proto-ETS site. These two have previously been suggested to be drivers, which is reinforced by our results supporting that positive selection is acting on these DNVs[2,3,17]. Their low frequency may be explained by a general scarcity of DNVs in most cancers; a

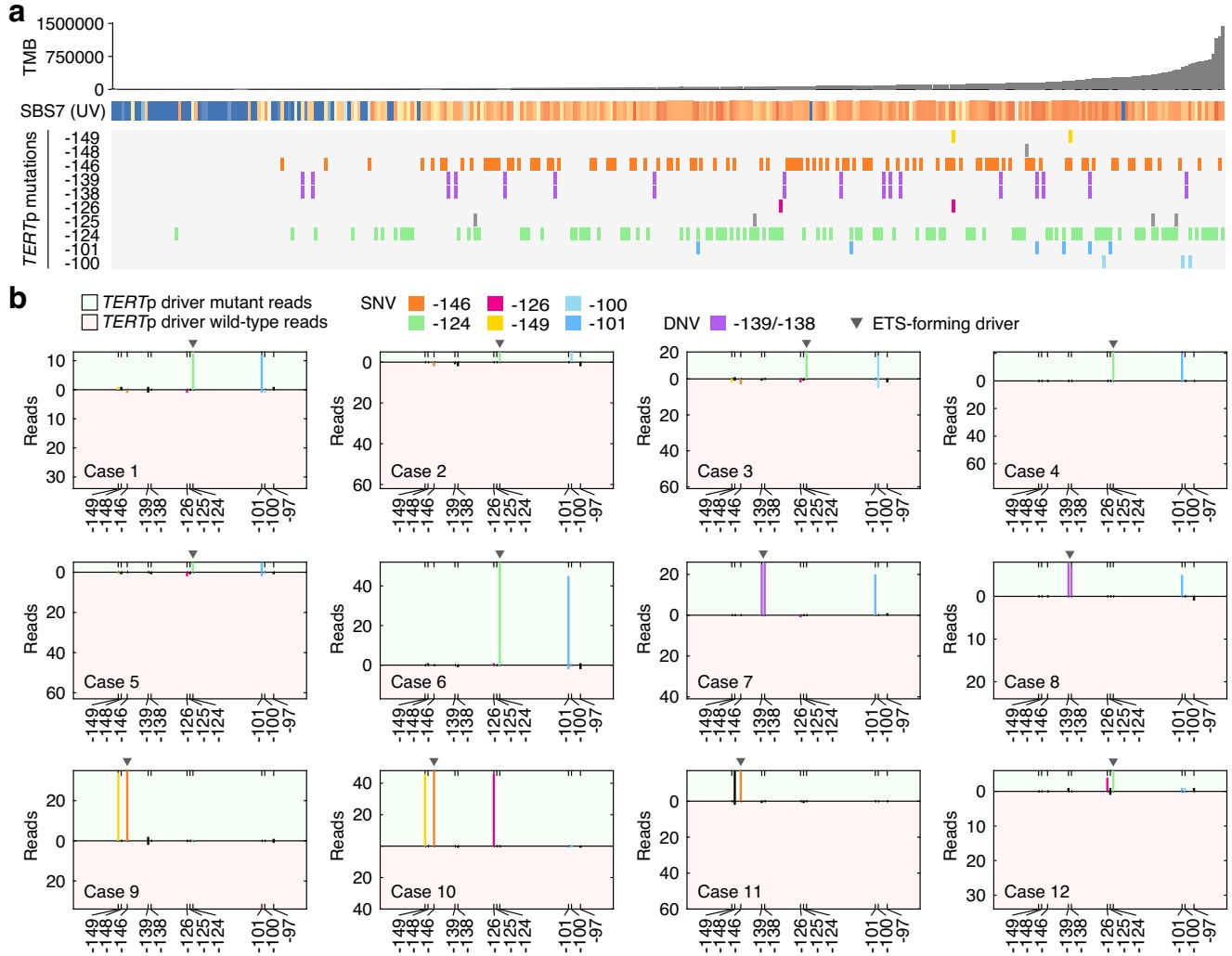

**Fig. 4 | Atypical *TERT*p mutations co-occur in cis with canonical drivers in 100k Genomes melanomas. a** Somatic mutations at positions of interest in the *TERT*p in the Genomics England 100k Genomes melanoma cohort. 335 samples are shown ordered by tumour mutation burden (TMB). The fraction of mutations attributed to the UV signature SBS7 is indicated for each sample (blue = 0%; orange = 100%). **b** Allele phasing in 12 cases with primary driver mutations (−146 bp, −139/−138 bp, or −124 bp) and atypical mutations. Reads were split by mutation status at the key

*TERT*p driver position, considering only reads covering all the relevant positions. The total number of driver mutant and wild-type reads are indicated by the height of the positive and negative axes, respectively. Mutations in cis with (i.e. in the same read as) the mutant allele or the reference allele are shown on the positive and negative axes, respectively. TMB, tumour mutation burden; SNV, single nucleotide variant; DNV, double nucleotide variant. Source data are provided as a Source Data file.

model supported the observation that the two events are near-exclusive to skin cancers, where UV-induced CC > TT DNVs are relatively abundant[31]. A small number of additional cases were found in bladder/urothelial tumours, indeed known to harbour CC > TT substitutions stemming from APOBEC mutagenesis (DBS11)[31].

The third category, and main focus of this study, encompasses non-ETS forming recurrent *TERT*p mutations (mainly at −149, −126, −101 and −100 bp), which all exhibited mutational patterns compatible with neutral selection thus deviating clearly from the ETS-forming events. Through genomic analyses of human tumours and in vitro CPD damage and mutagenesis studies, we show that these atypical events stem from UV-hypersensitive bases upstream of ETS-bound sites (CCTTCCK, underscored), whether pre-existing (native) or de novo-formed by driver mutations, further supporting that they are passengers. ETS-bound sites are known to exhibit weaker UV-hypersensitivity also at the central TC dipyrimidine[23]. While reflected in our CPD formation data as well as in UV mutagenesis results from cultured cells, mutations at this position were lacking in tumours. This is likely due to negative selection, as mutations at the centre of the core motif are

expected to counteract the primary driver event by disrupting ETS binding.

The native ETS site in the *TERT*p has been shown to act as a partner to de novo-formed sites at −146 and −124 bp, enabling recruitment of a multimeric ETS (GABP) complex to both sites[8,32]. There is strong support for a dependency on dual ETS sites for GABP complex formation and promoter activation including preferential site pair spacing that preserves the helical phase, both in the *TERT*p and in general[8,9]. While not recapitulated in our cell-free in vitro system, where recombinant GABP interacted also with a wild-type *TERT*p fragment via the native site, this model provides a plausible mechanism explaining a dependency on a primary driver event for native site mutations (at −101, −100 and −97 bp) in tumours. Notably, we also observed this dependency when inducing mutations in cultured melanoma cells by UV light. In further agreement with a dual sites model, in melanoma tumours we find that somatically mutated paired ETS sites in general exhibit preferential spacing reflecting the helical period as well as strong overlaps with GABP ChIP peaks (Supplementary Fig. 6). Given the mutant *TERT*p has been shown to undergo extensive

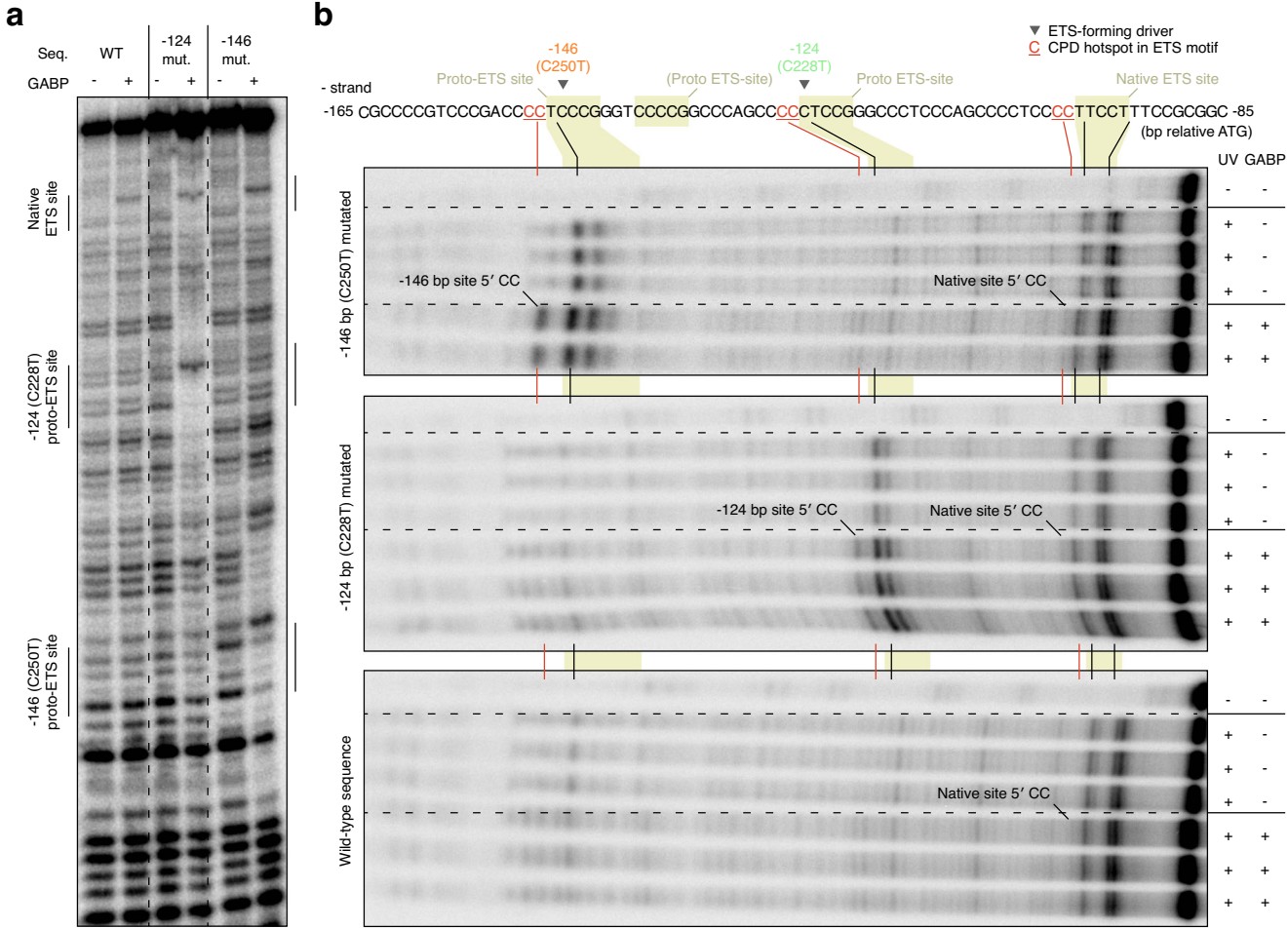

**Fig. 5 | Binding of GABP to the *TERT*p promotes UV damage formation at ETS hotspot sites in vitro. a** DNaseI footprint of wild-type or mutated radiolabelled *TERT*p dsDNA fragments encompassing the native and two main proto-ETS sites, with and without added recombinant GABPA/B protein complex. ETS sites are indicated and were localized as described in Supplementary Fig. 4. **b** CPD formation following UV irradiation (5000 J/m2 UVC) of the same *TERT*p fragments as revealed by T4 endonuclease V digestion, with and without GABP (see Supplementary Fig. 5 for additional gel image data). CPD, cyclobutane pyrimidine dimer. Source data are provided as a Source Data file.

chromatin remodelling[41] it is also plausible that the inactive wild-type allele is generally less accessible to regulatory proteins, which may contribute to an interaction between primary and native site mutations.

The native site is adjacent to a second pre-existing ETS site closer to the TSS, and of these tandem sites, the first (ETS−200) has been suggested to be the preferential GABP interaction partner[8,9]. It may be noted that this is consistently supported by our results, as no mutation hotspot effect was observed at the second site (ETS−195) neither in tumours nor after in vitro UV mutagenesis.

Dependency on a primary driver event is, in principle, mechanistically straight-forward in the case of atypical mutations at the two main proto-ETS sites. UV hotspot mutations at −149 bp and, rarely, at −148 bp (**CC**TTCCG) were indeed observed exclusively in conjunction with a −146 bp ETS-forming driver mutation that enables ETS docking and thus subsequent stimulation of UV damage (CPD) formation. The situation is analogous for hotspot mutations at −126 bp, which associated strongly with −124 bp ETS-forming driver mutations. However, in a few cases (4/14 in the GENIE), −126 bp SNVs were found to instead co-exist with −146 bp driver mutations. While mechanistically unclear, one may speculate that multimeric ETS complexes could still bind with sufficient affinity to this proto-ETS site following a −146 bp event, or that activation and remodelling of the −146 bp mutant *TERT*p allele allows another protein to bind and modulate damage at this coordinate.

The study of non-coding driver mutations has often been confounded by unexplained mutagenic phenomena[7], motivating careful deciphering of the origins of recurrent mutations in regulatory DNA. By clarifying the mechanism underlying atypical mutations in the *TERT*p, this study provides a more complete understanding of somatic alterations in one of the most frequently mutated regions in human cancer.

## Methods

### Processing of GENIE data for *TERT*p mutation analysis

Targeted sequencing data was obtained in tab-delimited Mutation Annotation Format (MAF) from GENIE v11 via the Synapse platform (http://synapse.org/genie). The initial dataset encompassed 118,094 tumour samples from 756 cancer subtypes (OncoTree codes) and 19 centres. For this study, samples sequenced with assays that lacked the *TERT*p region or that captured <300,000 bp were removed. For patients with multiple samples only the first reported sample was retained, thus ensuring that all samples were from unique donors. To enable robust calculation of UV mutation signature and mutation burden, cancer subtypes with <40 samples or <500 mutations in total in all samples were further disregarded. Finally, to focus only on cancer subtypes with *TERT*p mutations, cancer subtypes having <1% *TERT*p driver mutation frequency were removed. The final dataset comprised of 59 cancer types and 19,755 tumours (16.7% of GENIE v11) from Yale

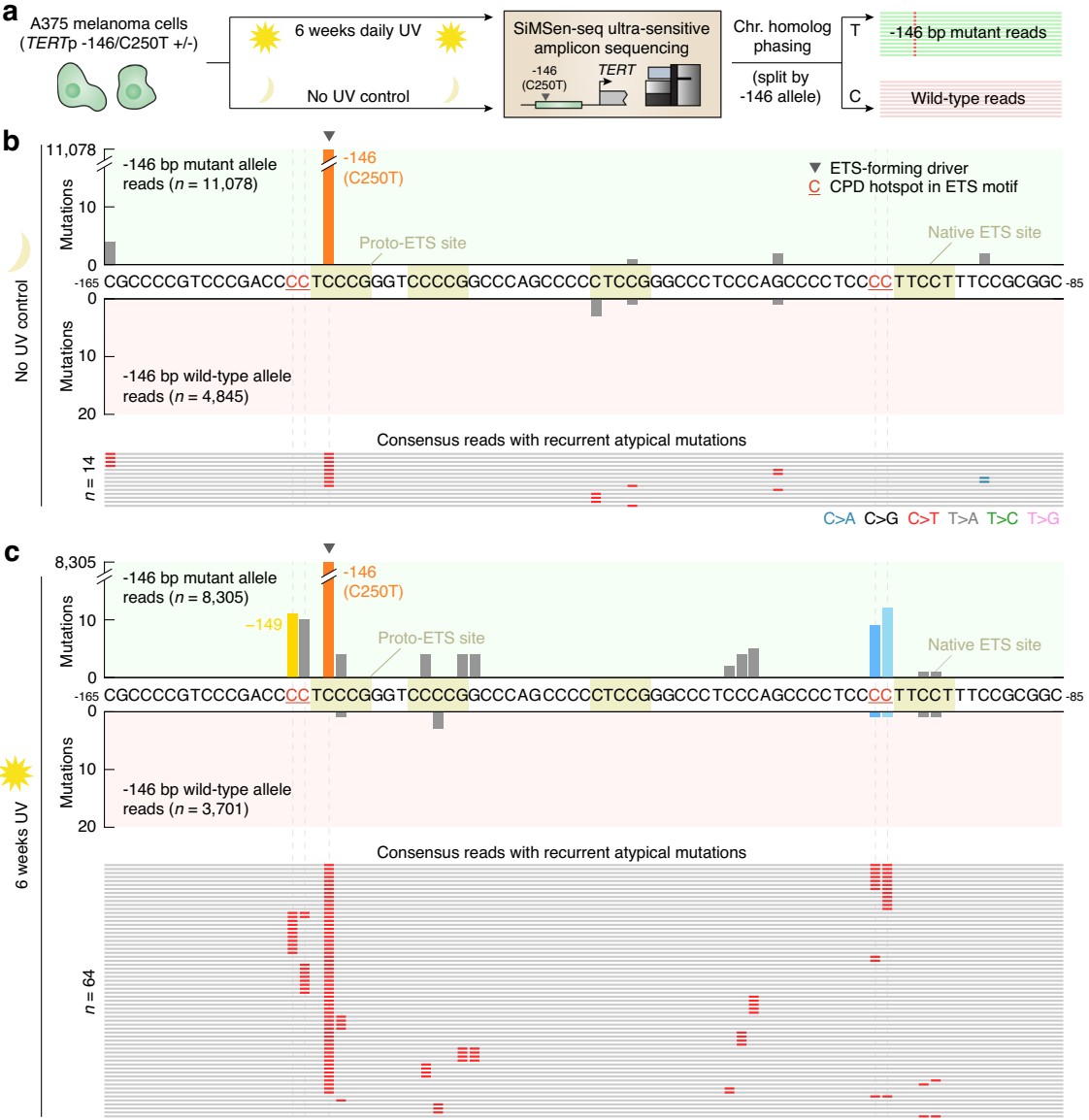

**Fig. 6 | UV exposure of A375 melanoma cells introduces mutations at predicted *TERT*p UV hotspot positions specifically on the −146 bp (C250T) mutated chromosome homologue. a** A375 cells, either untreated or UV-exposed (36 J/m2 UVC daily during 6 weeks), were assayed for subclonal *TERT*p mutations using a modified SiMSen-seq error-corrected amplicon sequencing protocol. A375 cells carry a −146 bp *TERT*p driver mutation, and mutations were phased in relation to this event by categorizing all error-corrected consensus reads (minimum 3 × oversampling) based on genotype at −146 bp (positive and negative axes). **b** Per-base mutation counts (same *TERT*p region as in Fig. 2) in non-exposed cells,

which had few recurrent mutations (15,923 consensus reads in total). All individual reads having recurrent mutations outside of the −146 site are shown. **c** Per-base mutation counts in UV-exposed cells (12,006 consensus reads), showing atypical recurrent mutations at predicted UV hotspot positions at the native ETS site (−101 and −100 bp) and at −149 and −148 bp, which are predicted secondary UV hotspot bases at the −146 bp proto-ETS site. The atypical mutations predominantly occurred in cis with the −146 bp driver mutation. Only mutations detected in >1 consensus read were considered. CPD, cyclobutane pyrimidine dimer. Source data are provided as a Source Data file.

(assay YALE-OCP-V3) and Memorial Sloan Kettering Cancer Center (assays MSK-IMPACT341, MSK-IMPACT410, MSK-IMPACT468) (Supplementary Table 2).

*TERT*p sites mutated in at least ten out of the 19,755 patients in the cohort were considered as recurrent in Fig. 1. As some single nucleotide variants (SNVs) were reported as being part of oligonucleotide variants (ONV), ONV were at this initial stage split into their corresponding SNP components, whereas dinucleotide variants (DNV) mutations were considered as a single events.

Single base substitution (SBS) and double base substitution (DBS) mutation signatures were calculated using the "DeconstructSigs" R package[42] (version 1.9.0 together with COSMIC SBS signatures version 2 and DBS signatures version 3) using a maximum of 5 signatures and

with the "exome2genome" option, with remaining parameters set to their default values. Recurrent driver mutations (SNVs present in ≥ 4 samples and DNVs present in more ≥ 2 samples in a given cohort) were removed before the analysis. To enable relevant observations about the role of UV light in the formation of *TERT*p mutations, cancer subtypes were classified into subcohorts based on the frequency of *TERT*p driver mutations (−124, −139/−138, −146 bp) in the samples and the fraction of mutations attributed to the UV mutation signatures SBS7 and DBS1. Specifically, the subcohorts were "High *TERT*p, UV" (≥20% of samples having *TERT*p driver mutations and ≥ 10% of all mutations being UV-attributed), "High *TERT*p, no UV" (≥20% *TERT*p drivers and < 10% UV mutations) and "Low *TERT*p" (<20% *TERT*p drivers). Per-sample UV mutation burdens in the "High *TERT*p, UV"

subcohort in Fig. 3 were calculated by considering C > T substitutions in dipyrimidine contexts, following removal of recurrent driver mutations as describe above. *P*-values for co-occurrence of atypical and driver mutations in patients were calculated using the two-sided Fisher's exact test without correction for multiple comparisons.

For several samples sequenced using the MSK-IMPACT assays, an error was identified in the GENIE database that incorrectly annotated a subset of −146 bp (C250T) mutations as ONVs together with −150 bp substitutions, confirmed by MSK (Prof. Michael Berger, personal communication). These artifacts were corrected for the data used in our analysis.

### Processing of Genomics England data for *TERT*p mutation analysis

Whole genome sequencing data from Genomics England was accessed through the Genomics England Research Environment. Patients clinically diagnosed with malignant melanoma (345 primary tumours) were initially selected using Genomics England LabKey database. Patients with multiple samples were filtered out, leaving 335 unique primary tumours for further processing. Alignment files (BAM format) and variant call files (VCF) were collected for all samples. Tumour mutation burdens were determined based on the VCF data, and per-sample mutational signature information was extracted using LabKey. Mutations at the key positions of interest (−97, −100, −101, −124, −125, −126, −138, −139, −146, −148 and −149 bp) were initially identified and visualized based on VCF mutations calls. In samples where atypical mutations co-occurred with ETS driver mutations, the events were phased based on the raw alignment data. Briefly, BAM files were uncompressed into SAM format using Samtools[43] while only maintaining reads that covered all the positions of interest indicated above. Mutations were called at each such position in each individual read by pinpointing all deviations from the reference sequence in the SAM data. For each sample, the reads were subsequently split by mutation status at the main driver positions, and mutation frequencies at all positions of interest were visualized separately for the two subsets.

### CPD profiling and DNase footprinting and EMSA

To create double stranded template, 90 bp oligos containing both the native ETS site and proto-ETS site at −124 and −146 bp in the *TERT*p (Supplementary Table 5) were labelled with γ-$^{32}$P. 100 pmol of forward oligo (C-rich strand) was incubated with 5 units of T4 PNK (Cat. No. M0201L, NEB) and 30 uCi of γ-$^{32}$P-ATP (Cat. No. SRP−301, Hartmann Analytic GmbH) in PNK buffer (70 mM Tris HCl, 10 mM MgCl$_2$, 5 mM DTT pH 7.6) in 50 ul for 60 min at 37 °C and heat inactivated for 20 min at 95 °C and cleaned up with a G25 column (Cat. No. 27532501, Cytiva). 100 pmol of reverse oligo (G-rich strand) was annealed to the forward oligo in a total volume of 100 ul in annealing buffer (15 mM Tris, 100 mM NaCl), incubated at 95 °C for 5 min and the heat block switched off and left overnight.

Recombinant GABPA/B heterodimer complex was produced by cloning genes for GABP β1L and GABP α, the former with an N-terminal His-tag, into the pET-Duet-1 vector followed by expression in Rosetta DE3 *E. coli* cells[9]. To determine the appropriate concentration of GABP protein complex to use in the UV footprinting experiment, we performed electrophoretic mobility shift assays (EMSA). GABP protein was diluted from 80 μM stock solution (final conc: 10 μM, 5 μM, 2.5 μM, 1 μM, 0.5 μM and 0.1 μM) and incubated with 20 nM labelled oligo in 15 μl reaction buffer (20 mM Tris-Cl (pH 7.5), 5% glycerol, 5.2 mM MgCl2, 50 mM KCl, 1 mM TCEP) for 1 h at room temperature and run on a 4% native PAGE gel in 0.5x TBE at 4 °C and imaged on a Typhoon scanner after 1 h exposure.

To perform CPD footprinting, 20 nM annealed oligo was incubated for 1 h at room temperature with or without 5 μM GABP protein in 15 μl reaction buffer in triplicate. 3 μl of this reaction was run on a 4% native PAGE gel in 0.5x TBE at 4 °C to confirm binding by EMSA and the

remaining 12 μl was irradiated with 5000 J/m2 UVC as a drop on parafilm. Irradiated DNA was diluted to 100 μl in H2O and subjected to phenol chloroform extraction followed by ethanol precipitation to remove GABP protein. DNA was resuspended in 15 μl T4 endonuclease V buffer (100 μg/μl BSA, 100 mM NaCl, 1 mM DTT, 1 mM EDTA, 25 mM Na$_2$HPO$_4$ pH 7.2) and cleaved at CPDs by incubating for 1 hr at 37 °C with 1 unit of T4 endonuclease V (Cat. No. M0308S, NEB). The reaction was stopped with the addition of 2x formamide buffer. Samples were loaded in appropriate amounts such that uncut oligo density, as determined using Multi Gauge v3.0 (Fujifilm), was similar across the wells. Samples were heated to 95% before running on an 8% native PAGE sequencing gel pre-heated to 55 °C and run for 1 h at 1500 V in 1x TBE. Cleavage products were visualized using a Typhoon imager with overnight exposure. For DNAse footprinting, 1 U DNAse I (Cat. No. EN0521, Thermo Fisher) was added to 20 nM oligo with or without GABP protein in reaction buffer and incubated for 5 min at RT. The reaction was stopped with the addition of 2x formamide buffer and 5 ul resolved on native PAGE 8% sequencing gels. Gel bands were quantified with ImageJ using the Analyze tool[44]. Uncropped gel images are included as Source Data.

### Ultrasensitive mutation analysis

A375 melanoma cells (a gift from the Jonas Nilsson laboratory, University of Gothenburg), which have a heterozygous *TERT*p mutation at −146 (C250T), were grown in DMEM + 10% FCS + gentamycin (Cat. No. 11965092, A5670701 and 15710064, respectively, Thermo Scientific). Cells were treated with 36 J/m2 UVC (254 nm) using a CL−1000 UV crosslinker (UVP) in DMEM in 10 cm plates without lids, 5 days a week for 6 weeks. Cells were reseeded at 1:5 density when confluent, and subsequently frozen at −20 °C. DNA was extracted using the DNeasy Blood and Tissue Kit (Cat. No. 69504, Qiagen). To identify subclonal mutations in the *TERT*p which arose following UV treatment, primers were designed to sequence the region using SiMSen-seq[37] (Supplementary Table 6). The high GC content required optimization of polymerases for amplification of *TERT*p. Barcoding of 100 ng genomic DNA and amplification proceeded as previously described but with Phusion Plus Mastermix and GC enhancer (Cat. No. F631S, Thermo Fisher). Sequencing was performed on an Illumina Novaseq 6000 instrument in 150 bp single end mode.

The fastq files containing the UMI-barcoded reads were preprocessed and subsequently aligned to the human reference genome (hg38) with UMIErrorCorrect[45]. Only consensus (error-corrected) reads based on at least three raw reads were maintained and indel-containing consensus reads were removed. Mutations that were present in more than one consensus read were considered in the final analysis.

### Analysis of somatic mutations at paired ETS sites

Whole genome somatic mutation calls from 183 melanoma patients from the MELA-AU project[46] was downloaded from the International Cancer Genome Consortium's (ICGC) database (http://dcc.icgc.org). Chromosomes X and Y were excluded from the analysis and only unique calls were considered for each donor. Matches to the GABP/ETS motif (NNTTCCK or NNTTCCG) were mapped in 1 kb upstream regions, defined based on NCBI RefSeq genes and using the 5′-most coding transcript for each gene. Pairs of sites on the same strand and with a start-to-start distance between 7 and 100 bp were retained for further analysis, thus avoiding overlapping sites. Site pairs were subdivided based on the number of overlapping mutations (non-mutated, ≥ 1 mutation/site and ≥ 2 mutations/site). Distance distributions for the different sets were plotted, applying a moving-average smoothening with a window size of 3 bp before normalization (division by total counts). The proportions of site pairs that overlapped with strong GABP ChIP-seq peaks (both site starts within the peak) were computed using ChIP-seq data for 8 cell lines (A549, GM12878, HeLa-S3, HepG2,

HL-60, K562, MCF-7 and SK-N-SH) from ENCODE[47] available via the UCSC browser in the Encode Haib Tfbs track, using the broadPeak files. Site pairs were required to overlap with at least one peak in one cell line and replicate, and only the top quartile of peaks (based on enrichment score, column 7 in the ENCODE files) were considered for each experiment.

## Reporting summary

Further information on research design is available in the Nature Portfolio Reporting Summary linked to this article.

## Data availability

Sequencing data generated for this study has been deposited in NCBI Sequence Read Archive (SRA) with BioProject accession number PRJNA1062776. Melanoma mutation data from the Australian Melanoma Genome Project via the ICGC data portal (https://docs.icgc-argo.org/docs/data-access/icgc-25k-data). NCBI RefSeqAll gene annotations for hg19, via the UCSC table browser (https://genome.ucsc.edu/cgi-bin/hgTables). ENCODE ChIP-seq data for 8 cell lines (A549, GM12878, HeLa-S3, HepG2, HL-60, K562, MCF-7 and SK-N-SH) from UCSC. GENIE v11 (https://www.synapse.org/#!Synapse:syn26706564). Genomics England (https://www.genomicsengland.co.uk/research/research-environment). Source data are provided with this paper.

## Code availability

Code is provided as a Supplementary Data 1.

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

## Acknowledgements

The authors would like to acknowledge the American Association for Cancer Research and its financial and material support in the development of the AACR Project GENIE registry, as well as members of the consortium for their commitment to data sharing. This research was made possible through access to data in the National Genomic Research Library, which is managed by Genomics England Limited (a wholly owned company of the Department of Health and Social Care). The National Genomic Research Library holds data provided by patients and collected by the NHS as part of their care and data collected as part of their participation in research. The National Genomic Research Library is funded by the National Institute for Health Research and NHS England. The Wellcome Trust, Cancer Research UK and the Medical Research Council have also funded research infrastructure. Jesslyn E. Park and Dr. Gavin Knott are acknowledged for their assistance in optimizing GABP expression and purification. E.L. was supported by grants from the Knut and Alice Wallenberg Foundation, the Swedish Medical Research Council and the Swedish Cancer Society. T.L and J.V.d.E was funded by Kom op tegen Kanker (Stand up to Cancer) and the Flemish Cancer Society. M.F. was funded by the Swedish Cancer Society and the Knut and Alice Wallenberg Foundation.

## Author contributions

E.L. and K.E. designed the study. K.E., K.M.S and E.H. performed experiments. V.K.S., A.B., M.L., L.Ö., T.L., K.E. and E.L. analysed data. A.S., M.F., J.D. and J.V.d.E acted in a supervisory capacity. K.E. and E.L. wrote the manuscript.

## Funding

## Competing interests

A.S. is a co-inventor of the SiMSen-Seq technology that is patent protected (U.S. Serial No.:15/552,618). A.S. declares stock ownership in Tulebovaasta, Iscaff Pharma and SiMSen Diagnostics, and is a board member of Tulebovaasta. The other authors declare no competing interests.
