## [Transparent Peer Review file · Nature Communications]

Mechanistic basis of atypical TERT promoter mutations

Corresponding Author: Professor Erik Larsson

Version 0:

Reviewer comments:

Reviewer #1

(Remarks to the Author)

It has previously been shown by Larsson et al. (and other groups), that ETS binding sites are strong mutation hotspots due to their propensity for ETS binding to induce increased UV-induced DNA damage. On the back of this, in this study, Larsson and colleagues test the hypothesis that the creation of driver ETS binding sites at TERT promoters may lead to the formation of secondary atypical TERTp mutations. The study clearly validates the hypothesis using a combination of analysis of existing cancer genomics data with in-cell validation. The study is a very nice illustration of mutational processes in cancer genomes. I only have a few comments:

1. In Figure 1, it is not clear how samples/patients with more than one TERTp mutation are represented. Presumably, they can only be counted once since the TERTp mutation fraction does not exceed 1. If this is the case, which co-occurring TERTp mutation is indicated in the bar?
2. Would it be possible to additionally use mutation allele frequency (MAF) to show that the primary TERTp mutations have higher MAF compared with the atypical TERTp mutation?
3. The heading "Phasing in 100k Genomes molecularly links atypical TERTp mutations to primary drivers" is a little misleading as it sounds like 100k genomes are being analysed. It would be better say something like "Phasing of melanoma genomes from the 100k Genomes cohort..."
4. It would be nice to demonstrate that the atypical mutation no longer occurs in the GABPA/B knocked out A375 cells. It is probably out of scope of this study, but interesting to know if it has been attempted.
5. Does the results imply that the atypical TERTp mutations do not affect GABP binding? And would this also provide further support that ETS binding site hotspot mutations are generally passengers?

Reviewer #2

(Remarks to the Author)

TERT promoter mutations are among the most prevalent driver mutations in human cancers. While there are two hotspot canonical TERT promoter mutations, many other variants have not been characterized. Among these are mutations that co-occur with hotspot driver mutations. The authors hypothesize that the collateral, atypical mutations are subclonal passenger mutations that are acquired after the canonical hotspot mutations, that they are induced by UV damage, and that ETS factor binding to the de novo and native sites explains why they occur where they do. The authors' in silico analyses and in vitro experiments support the hypotheses. However, it is not as clear whether the findings represent a significant conceptual advance. The interplay between ETS transcription factor binding and formation of UV-associated cyclobutane pyrimidine dimers has been reported, and here it is applied in the context of the mutant TERT promoter. The data support the interplay in the mutant TERT promoter in tumor types with high UV exposure. Therefore it seems important but novelty is moderate. The clinical and biomedical utility/implications of these findings, if any, should be described because the atypical mutations are passengers, and thus have no role in TERT activation or tumor immortality. Are the authors simply saying they are passengers and in the clinical setting they can be described as such? Given they likely occur after the driver mutations, it seems logical. On the other hand, the authors make an interesting and somewhat novel observation: In figure 2B and 6C, in the context of the mutant TERT promoter, the authors suggest that the ETS-200 site is preferentially bound rather than the ETS-195 site. To our knowledge, this would be the first evidence of site preference derived from the endogenous locus in patient samples.

Minor points:

- A note on sub-section heading “Phasing in 100k Genomes molecularly links atypical TERTp mutations to primary drivers.” We are not certain that phasing is the correct term here as the authors only identify that the driver and potential passenger mutations occur on the same allele.
- Use of “phase” in figure 4 legend and related text in manuscript: we recommend changing the text from “phase-linked” and the like to “cis”.
- Figure 2, 5, 6, it is difficult to see the CPD hotspot as a red C.
- The authors state that “By clarifying the role of most recurrent mutations in the TERTp, the results from this study will facilitate future interpretation of somatic alternations in one of the most frequently mutated genes in human cancer.” However, the manuscript does not test nor clarify the role of the hotspot mutations. Also, the authors do not present data related to the effect, or lack thereof, of these atypical mutations on TERT expression. It may not be worthwhile proving a negative, however. Since the atypical mutations are subclonal, rare passengers that occur following and in cis with the canonical hotspot TERTp mutation, it is not clear if/how they would be incorporated into the interpretation of TERTp alterations.

Reviewer #3

(Remarks to the Author)

The manuscript by Elliott et al., builds upon previous works from the authors, now focusing on the characterisation of the formation of atypical --albeit recurrent at low frequencies-- TERT promoter (TERTp) mutations in skin cancers. Through the analysis of somatic mutations from thousands of cancer samples, authors show atypical TERTp mutations are more frequently found in cohorts with TERT mutations and UV-light exposure compared to those with TERT mutations and low UV-light exposure. In this scenario, authors find that atypical TERTp mutations are located in the vicinity of ETS motifs and they often co-occur with driver TERTp events (-146/C250T and -124/C228T) at ETS sites. These observations lead the authors to suggest a two-hit model, where the atypical mutations arise as a result of UV-light and de novo ETS binding sites created by driver mutations. They carry out experimental work to support this hypothesis, first showing an increased formation of UV-light damage overlapping atypical TERTp mutated positions upon ETS binding. Finally, they demonstrate atypical TERTp mutations form in the vicinity of ETS sites (native or de novo at -146/C250T) after UV exposure.

While the interaction of UV-light, transcription factor binding (including ETS family) and DNA repair is known to cause passenger mutational hotspots in upstream regulatory regions, this is, to the best of my knowledge, the first detailed analysis on the mutagenic mechanisms acting on TERTp. In the context of skin cancers, previous work by the authors and others have shown that recurrent mutations in promoter sequences are a result of mutational processes rather than selection, with TERTp being a notable exception. The current manuscript by Elliott et al. refines these observations and contributes to better understanding the mutational landscape in TERTp. From my perspective, considering that TERT alterations are among the top driver events across cancers, this is a relevant question to pursue.

There are, however, some major questions that need to be addressed prior to publication, as detailed below.

Major

- 1) Mutations arise from the interaction between DNA damage and DNA repair. While the authors provide convincing data supporting a role for UV-light damage in the formation of TERTp atypical mutations, the effect of DNA repair has not been tested. The authors conclusions need to be supported by additional data exploring the role of DNA repair, similarly as they have shown in previous works (for example, Elliot et al., 2018). The main question to be addressed is whether TERTp atypical mutations are formed in the absence/presence of global and/or transcription-coupled nucleotide excision repair (NER). More specifically, are these mutations formed in UV-exposed A375 cells with C250T TERTp mutation after NER knockdown? Likewise, do these mutations occur in NER deficient skin cancers from GENIE, 100k Genomes, or other cancer sequencing cohorts?
- 2) Definition of low TERT + UV group. This group, defined as <20% TERTp drivers and $\geq 10\%$ SBS7 mutations, is mostly composed of non-skin cancers, yet it does not seem biologically plausible that the fraction of mutations attributed to UV-light signatures SBS7 and DBS1 (middle and right panels in Figure 1) in these samples are similar to those in skin cancers within the high TERT + UV group. How is this explained? Could this be driven by particular samples? Can authors provide additional data to check the accuracy of the signature fitting and discard any potential artefact?
- 3) Figure 3. These plots aim to show that the frequency distributions of driver and atypical hotspot mutations versus mutation burden are different. Can authors provide any statistics to support this claim (the text refers to correlation, but this is not formally tested)? Also, does the mutation burden shown here include total mutations in each group --this is, UV-light and non-UV light caused mutations--? Considering that the authors' hypothesis is that atypical mutations are caused by UV-light, it would be relevant to replicate this analysis using SBS7/DBS1 mutation burden alone.
- 4) Figure 5b. The results could be improved if the authors quantified and compared the differences in CPD signals across the experimental groups.
- 5) The methods describing the signature fitting need to provide additional details, such as the reference set of signatures used.
- 6) While the data to reproduce the computational analysis of GENIE samples is provided, the code does not seem to be

available for this or other computational analyses.

Minor

- 1) As an additional control, it would be interesting to show if there are differences in the frequencies of TERTp atypical mutations in high UV-light exposed and low/no UV-light-exposed skin melanomas.
- 2) In the second section of results, on page 5 line 130, it is mentioned that atypical mutations at -101, -100 and -97 add up to 44 samples, but according to Supplementary Table 3 this should be 24 samples.
- 3) Figure 1. Given the low frequency of atypical TERTp mutations compared to driver mutations, it is difficult to evaluate differences in the frequency of atypical mutations across the cohorts. Perhaps some changes in the colour codes, or plotting these mutations alone in a different panel, could help.
- 4) Figure 2 caption. TERT coordinates are written as chr8:1295188-1295268 where it should be chr5:1295188-1295268.
- 5) Figure 4b. Would it be possible to show the same scale on the y axis for the different sets of reads to better understand the frequencies between both alleles? For example, the y axis in Case 1 TERTp driver wild-type reads could range from 0 to 10, as it is shown on driver mutant reads.
- 6) Figure 6b is missing y axis labels.
- 7) Supplementary Table 1. There are different hg19 mutation coordinates that have been incorrectly mapped to hg38, including chr5:1295228.
- 8) Supplementary Table 3. The values of this table need to be reviewed and corrected. For example, counts and percentages shown for individual driver mutations do not add up to "All drivers" in any columns except in "Low TERTp no UV". Likewise, the values within "All native and -146" seem to be incorrect based on the rows above.

Version 1:

Reviewer comments:

Reviewer #1

(Remarks to the Author)

Thank you for addressing all my concerns.

Reviewer #2

(Remarks to the Author)

The authors have revised the manuscript to address my critique in full.

Reviewer #3

(Remarks to the Author)

I wish to thank the authors for their work and the clear and comprehensive answers to my comments. The revisions have addressed all the points raised in my first review, and I believe the new version of the manuscript has improved.

I appreciate the author's explanation on the limitations to show the role of DNA repair in this model and within cancer DNA-sequencing cohorts available to date. I believe this information would be valuable to other readers, so I wonder whether it could be incorporated into the manuscript. I would leave this decision to the authors' discretion.

From my point of view, the manuscript can be published in its current form. Congratulations again to the authors on this work.

Point by point response - NCOMMS-24-33636
Mechanistic basis of atypical *TERT* promoter mutations

We wish to sincerely thank all reviewers for their efforts in reviewing our manuscript. The manuscript has been revised as detailed below, where reviewer comments are shown in black, and our responses are indicated in blue.

Additional changes:

- Source Data tables have been added for all figures, including those in the supplement.
- Complete source code has been included as a zip archive.
- Samples based on the smaller COLU capture assay in GENIE have been removed from **Fig. 1**. This assay has limited sequence coverage and was previously excluded in some downstream analyses, and it therefore made sense to remove the relevant samples right from the start. While this has negligible impact on figures and numbers, all relevant figures, tables and numbers have been updated to account for minor changes in the base mutation dataset.
- Various improvements to text clarity.

REVIEWER COMMENTS

Reviewer #1 (Remarks to the Author):

It has previously been shown by Larsson et al. (and other groups), that ETS binding sites are strong mutation hotspots due to their propensity for ETS binding to induce increased UV-induced DNA damage. On the back of this, in this study, Larsson and colleagues test the hypothesis that the creation of driver ETS binding sites at *TERT* promoters may lead to the formation of secondary atypical *TERT*_p mutations. The study clearly validates the hypothesis using a combination of analysis of existing cancer genomics data with in-cell validation. The study is a very nice illustration of mutational processes in cancer genomes. I only have a few comments:

We thank the reviewer for these positive comments.

1.1. In Figure 1, it is not clear how samples/patients with more than one *TERT*_p mutation are represented. Presumably, they can only be counted once since the *TERT*_p mutation fraction does not exceed 1. If this is the case, which co-occurring *TERT*_p mutation is indicated in the bar?

We thank the reviewer for pointing this out. The bar indicated the fraction of samples carrying each mutation, regardless of co-occurrence with other mutations. As such it was in principle possible for the bar to exceed 1, although this doesn't happen in practice. However, we can only agree this was not clear.

In **Fig. 1**, we have now split up the canonical drivers and atypical mutations such that they are shown in separate graphs. As the mutations within each of these two sets are mutually exclusive (with a few very rare exceptions), the two axes now indicate the total fraction of samples carrying a primary driver mutation or an atypical mutation, respectively:

1.2. Would it be possible to additionally use mutation allele frequency (MAF) to show that the primary TERTp mutations have higher MAF compared with the atypical TERTp mutation?

This is a relevant question, and while not previously addressed in the text, the information is available **Fig. 4**, which shows allele frequencies of primary and secondary TERTp mutations using Genomics England WGS data. As can be seen in this figure, primary/secondary frequencies are essentially the same, or only slightly lower for the atypical variants, supporting presence of both variants in an early expanded cell clone (excerpt from **Fig. 4** where triangles indicate primary drivers):

TERTp secondary variants in general have similar, or lower, allele frequencies compared to primary driver variants also in GENIE (see plot below):

However, we suspect that *TERTp* allele frequencies may not be fully reliable in GENIE. High GC content and strong secondary structure (a G4 element) makes the *TERTp* notoriously difficult to sequence (see e.g. PMID 33940787). G4 structures can impede the progression of DNA polymerase to different extents depending on the exact sequence, leading to skewed frequency estimates due to selective dropouts where one allele is favoured over the other (PMID 34650044). Being based on hybridization capture assays, if anything GENIE may be more sensitive to such effects. Notably, Genomics England uses a PCR-free protocol (PMID 38200255), which is likely favourable in this context.

We have added the following text to Results to highlight VAF data from the Genomics England melanoma cohort (row 186):

“Compared to primary mutations, secondary events had similar or only slightly lower allele frequencies, supporting that they occurred early before expansion of major cell clones (Fig. 4b)”.

1.3. The heading “Phasing in 100k Genomes molecularly links atypical *TERTp* mutations to primary drivers” is a little misleading as it sounds like 100k genomes are being analysed. It would be better say something like “Phasing of melanoma genomes from the 100k Genomes cohort...”

Thank you for pointing this out. We have modified the title to the following, not mentioning 100k Genomes to avoid making it too long (row 164):

*“Allele phasing molecularly links atypical *TERTp* mutations to primary drivers”*

1.4. It would be nice to demonstrate that the atypical mutation no longer occurs in the GABPA/B knocked out A375 cells. It is probably out of scope of this study, but interesting to know if it has been attempted.

While interesting in principle, in practice this will be challenging to test and we would have to agree it seems out of the scope of the study: while GABPA/B is the main ETS complex suggested to bind to the *TERT*_p, there is also a proposed role for ETS1/p52 (e.g. Xu et al, PMID 30093619), which may thus provide redundancy in case of GABP inactivation. Further complicating things, there are 28 different ETS proteins in humans with considerable similarity in DNA sequence specificity across the family. It may be noted that we do show dependency on the GABP complex in vitro in the absence of other ETS factors (**Fig. 5**).

1.5. Does the results imply that the atypical *TERT*_p mutations do not affect GABP binding? And would this also provide further support that ETS binding site hotspot mutations are generally passengers?

This interpretation is correct. The atypical mutations arise at bases outside of the ETS recognition motif, TT \overline{C} CK, which lack importance for ETS binding to DNA. This question also puts the spotlight on an interesting observation which we previously failed to highlight in the manuscript:

The in vitro UV mutagenesis experiment (**Fig. 6**) shows some mutations arising also at the central TT \overline{C} CK position, which is in fact expected since there is a CPD hotspot effect also at the central TC dipyrimidine of ETS-bound sites, in addition to 5' flanking bases (supported by our CPD profiling data in **Fig. 5** as well as prior results e.g. PMID 28489852). These mutations *will* disrupt the core motif and thus ETS binding - but they are not seen in the tumour data (GENIE or 100k Genomes). This discrepancy is very likely explained by negative selection in the tumours: since there is selective pressure for the *TERT*_p driver mutations, there should also be selection against nearby cis variants that directly counteract their effect.

We have now made note of this in the discussion (row 254):

“ETS-bound sites are known to exhibit weaker UV-hypersensitivity also at the central TC dipyrimidine. While reflected in our CPD formation data as well as in UV mutagenesis results from cultured cells, mutations at this position were lacking in tumours. This is likely due to negative selection, as mutations at the centre of the core motif will counteract the primary driver event by disrupting ETS binding.”

We have also further underscored that the atypical mutations are passengers (row 161):

*“These results are consistent with ETS-forming *TERT*_p mutations being drivers and with non-ETS-forming atypical events arising passively due to UV exposure thus being passengers.”*

And (row 251):

“Through genomic analyses of human tumours and in vitro CPD damage and mutagenesis studies, we show that these atypical events stem from UV-hypersensitive bases upstream of ETS-bound sites (CCTTCCK, underscored), whether pre-existing (native) or de novo-formed by driver mutations, further supporting that they are passengers.”

Reviewer #2 (Remarks to the Author):

2.1. TERT promoter mutations are among the most prevalent driver mutations in human cancers. While there are two hotspot canonical TERT promoter mutations, many other variants have not been characterized. Among these are mutations that co-occur with hotspot driver mutations. The authors hypothesize that the collateral, atypical mutations are subclonal passenger mutations that are acquired after the canonical hotspot mutations, that they are induced by UV damage, and that ETS factor binding to the de novo and native sites explains why they occur where they do. The authors' in silico analyses and in vitro experiments support the hypotheses. However, it is not as clear whether the findings represent a significant conceptual advance. The interplay between ETS transcription factor binding and formation of UV-associated cyclobutane pyrimidine dimers has been reported, and here it is applied in the context of the mutant TERT promoter. The data support the interplay in the mutant TERT promoter in tumor types with high UV exposure. Therefore it seems important but novelty is moderate. The clinical and biomedical utility/implications of these findings, if any, should be described because the atypical mutations are passengers, and thus have no role in TERT activation or tumor immortality. Are the authors simply saying they are passengers and in the clinical setting they can be described as such? Given they likely occur after the driver mutations, it seems logical.

Given that the *TERT*_p -124 and -146 bp positions are among the most frequently mutated in all human cancer (only KRAS G12 ranks above the -124 bp *TERT*_p mutations pan-cancer in PCAWG; see Rheinbay et al, PMID 32025015), we think it is relevant to sort out the role and origin of the additional mutations identified in this region. As noted, our conclusion is that they are passengers that arise “passively” due to a specific localized mutational process, explaining why they are restricted to UV-exposed tumours. To ensure this is clear, we have further underscored in the manuscript that atypical mutations are passengers (row 161):

*“These results are consistent with ETS-forming *TERT*_p mutations being drivers and with non-ETS-forming atypical events arising passively due to UV exposure thus being passengers.”*

And (row 251):

“Through genomic analyses of human tumours and in vitro CPD damage and mutagenesis studies, we show that these atypical events stem from UV-hypersensitive bases upstream of ETS-bound sites (CCTTCCK, underscored), whether pre-existing (native) or de novo-formed by driver mutations, further supporting that they are passengers.”

2.2. On the other hand, the authors make an interesting and somewhat novel observation: In figure 2B and 6C, in the context of the mutant TERT promoter, the authors suggest that the ETS-200 site is preferentially bound rather than the ETS-195 site. To our knowledge, this would be the first evidence of site preference derived from the endogenous locus in patient samples.

This is correct, and as noted by the reviewer this is otherwise quite difficult to show in patient samples, although earlier published results do point in the same direction (Bell, Science 2015, PMID 25977370; Barger, Nature Commun 2022, PMID 36114166). Not only is a UV hotspot effect visible only at the ETS-200 site in tumours, but our in vitro UV mutagenesis experiment (**Fig. 6**) clearly supports this too. We have made note of this in the discussion, now referring to the previously used ETS-200/195 nomenclature for clarity (row 275):

“The native site is adjacent to a second pre-existing ETS site closer to the TSS, and of these tandem sites, the first (sometimes referred to as ETS-200) has been suggested to be the preferential GABP interaction partner^{8,9}. It may be noted that this is consistently supported by our results, as no mutation hotspot effect was observed at the second site (ETS-195) neither in tumours nor after in vitro UV mutagenesis.”

Minor points:

2.3. A note on sub-section heading “Phasing in 100k Genomes molecularly links atypical TERTp mutations to primary drivers.” We are not certain that phasing is the correct term here as the authors only identify that the driver and potential passenger mutations occur on the same allele. Use of “phase” in figure 4 legend and related text in manuscript: we recommend changing the text from “phase-linked” and the like to “cis”.

We thank the reviewer for pointing out that the terminology was not clear, and as suggested we have changed the relevant Results section as follows (row 183):

*“We found that mutations at the native ETS site were always in cis with a primary mutation thus having arisen on the same chromosome homolog (**Fig. 4b**). Similarly, atypical mutations at -149 bp and -126 were in cis with -146 bp and -124 bp primary driver mutations, respectively.”*

Similar changes were also done in the figure legend and the supplement, as well as in relation to **Fig. 6**, in which phasing of in vitro-induced UV mutations was performed.

Note that we still make use of the term “phasing”, which we believe should correctly describe a procedure that determines if alleles are on the same chromosome homolog or not.

2.4. Figure 2, 5, 6, it is difficult to see the CPD hotspot as a red C.

Thank you for pointing this out. We have now underscored these positions to make them stand out better, which hopefully rectifies the problem:

2.5. The authors state that “By clarifying the role of most recurrent mutations in the *TERT*_p, the results from this study will facilitate future interpretation of somatic alternations in one of the most frequently mutated genes in human cancer.” However, the manuscript does not test nor clarify the role of the hotspot mutations. Also, the authors do not present data related to the effect, or lack thereof, of these atypical mutations on *TERT* expression. It may not be worthwhile proving a negative, however. Since the atypical mutations are subclonal, rare passengers that occur following and in cis with the canonical hotspot *TERT*_p mutation, it is not clear if/how they would be incorporated into the interpretation of *TERT*_p alterations.

We agree that the manuscript may be concluded in a better way and thank the reviewer for pointing this out. What we wanted to convey was that, with these results we now have a good understanding of essentially all mutations that are known to occur in the *TERT*_p: there is additional support that rare ETS-forming DNVs (-139/-138) are truly drivers, and a clear explanation of the mechanism behind the non-ETS-forming atypical variants, strongly supporting that they are passengers. We do think this adds something relevant to the body of knowledge about the *TERT*_p and mutations therein and how these should be interpreted.

In the final paragraph, “role” simply referred to a “passenger role”. However, based on this comment, we have in the end simplified this paragraph to the following (row 291):

*“The study of non-coding driver mutations has often been confounded by unexplained mutagenic phenomena⁷, motivating careful deciphering of the origins of recurrent mutations in regulatory DNA. By clarifying the mechanism underlying atypical mutations in the *TERT*_p, this study provides a more complete understanding of somatic alterations in one of the most frequently mutated regions in human cancer.”*

Reviewer #3 (Remarks to the Author):

The manuscript by Elliott et al., builds upon previous works from the authors, now focusing on the characterisation of the formation of atypical --albeit recurrent at low frequencies-- TERT promoter (TERTp) mutations in skin cancers. Through the analysis of somatic mutations from thousands of cancer samples, authors show atypical TERTp mutations are more frequently found in cohorts with TERT mutations and UV-light exposure compared to those with TERT mutations and low UV-light exposure. In this scenario, authors find that atypical TERTp mutations are located in the vicinity of ETS motifs and they often co-occur with driver TERTp events (-146/C250T and -124/C228T) at ETS sites. These observations lead the authors to suggest a two-hit model, where the atypical mutations arise as a result of UV-light and de novo ETS binding sites created by driver mutations. They carry out experimental work to support this hypothesis, first showing an increased formation of UV-light damage overlapping atypical TERTp mutated positions upon ETS binding. Finally, they demonstrate atypical TERTp mutations form in the vicinity of ETS sites (native or de novo at -146/C250T) after UV exposure.

While the interaction of UV-light, transcription factor binding (including ETS family) and DNA repair is known to cause passenger mutational hotspots in upstream regulatory regions, this is, to the best of my knowledge, the first detailed analysis on the mutagenic mechanisms acting on TERTp. In the context of skin cancers, previous work by the authors and others have shown that recurrent mutations in promoter sequences are a result of mutational processes rather than selection, with TERTp being a notable exception. The current manuscript by Elliott et al. refines these observations and contributes to better understanding the mutational landscape in TERTp. From my perspective, considering that TERT alterations are among the top driver events across cancers, this is a relevant question to pursue.

We were glad to receive these positive general remarks.

There are, however, some major questions that need to be addressed prior to publication, as detailed below.

Major

3.1. Mutations arise from the interaction between DNA damage and DNA repair. While the authors provide convincing data supporting a role for UV-light damage in the formation of TERTp atypical mutations, the effect of DNA repair has not been tested. The authors conclusions need to be supported by additional data exploring the role of DNA repair, similarly as they have shown in previous works (for example, Elliot et al., 2018). The main question to be addressed is whether TERTp atypical mutations are formed in the absence/presence of global and/or transcription-coupled nucleotide excision repair (NER). More specifically, are these mutations formed in UV-exposed A375 cells with C250T TERTp mutation after NER

knockdown? Likewise, do these mutations occur in NER deficient skin cancers from GENIE, 100k Genomes, or other cancer sequencing cohorts?

We think that the data we present in the manuscript, together with prior knowledge, leaves little doubt regarding the basic mechanism:

It is already established that the ETS hotspot effect per se does not rely on repair (or inhibition thereof), and even in XPC *-/-* (global NER deficient) tumours, ETS hotspots remain (Fredriksson 2017, PMID 28489852; Elliott et al 2018, PMID 30586386).

Specifically, for the *TERT* promoter, we show that actual damage (CPD formation) is elevated at the exact positions of interest, notably in an experimental system that lacks repair (**Fig. 5**). This damage is not repaired in cells (at least not sufficiently), since mutations then arise at precisely these sites (shown in **Fig. 6** in a controlled experimental system, in addition to the tumour data).

While it is possible that a reduction in NER efficacy known to occur around protein-bound sites in general (as shown in Sabarinathan et al, PMID 27075101, and Perera et al, PMID 27075100) could further add to the likelihood of mutagenesis, this would not change the conclusion regarding the basic mechanism or explain why mutations arise at these precise bases.

The *TERT*_p UV hotspots are in absolute terms less sensitive (thus requiring larger total UV dose) than top established ETS hotspots such as the *RPL13A* promoter site, which we have studied previously (e.g. Elliott et al, PLOS Genetics 2018). Even in wild-type A375 cells, we were required to perform our UV exposure experiment using low doses spread out over a long period of time (1.5 months) to avoid cell death while obtaining sufficient signal. NER deficiency renders cells considerably more vulnerable to UV (in the same way that human Xeroderma patients are UV-hypersensitive). Studying formation of the *TERT*_p secondary hotspots in a repair-deficient background would therefore be challenging, in addition to adding limited value compared to what is already presented in our opinion.

While it would in principle be straight-forward to study formation of secondary hotspots in sequencing data from NER-deficient skin tumours, a large cohort would be needed to have a chance to see them, and such data currently does not exist to our knowledge. Xeroderma (lacking NER) skin tumours are uncommon and not represented in the major consortia (100k Genomes, TCGA, GENIE). In Zheng et al (Cell Reports 2014), probably the most widely used XPC *-/-* skin tumour cohort (five skin squamous cell carcinomas), there are unfortunately no samples with primary *TERT*_p driver mutations.

3.2. Definition of low TERT + UV group. This group, defined as <20% TERT_p drivers and ≥10% SBS7 mutations, is mostly composed of non-skin cancers, yet it does not seem biologically plausible that the fraction of mutations attributed to UV-light signatures SBS7 and DBS1 (middle and right panels in Figure 1) in these samples are similar to those in skin cancers within the high

TERT + UV group. How is this explained? Could this be driven by particular samples? Can authors provide additional data to check the accuracy of the signature fitting and discard any potential artefact?

We thank the reviewer for pointing this out. We have now critically revised the UV signature burden analysis, which has been improved in several ways giving us added confidence in the results (**Fig. 1**).

The analysis is complicated by the use of small capture panels in GENIE, leading to low mutation counts and sometimes a high proportion of driver mutations, which may have an unproportional impact on the outcome. Therefore, we have now removed all recurrent mutations in each subcohort ($n \geq 4$ samples) thus eliminating common drivers (such as BRAF V600E) while retaining the informative passengers. We have also revised the parameters to DeconstructSigs as these were previously too inclusive leading to spurious low SBS7 detections in many cancers. Finally, we have altered the presentation such that we show SBS7 and DBS1 in the same stacked bar graph, indicating their relative contribution to all mutations (SBS+DBS).

However, we can conclude that there is still a solid UV signal in the subcohorts mentioned above, supported by consistent simultaneous SBS7 and DBS1. The reason for this is obvious in the case of Merkel cell carcinoma, but it is also likely that many of the CUPs (“cancers of unknown primary”) are skin samples, and the same may be true e.g. for SARCNOs (“sarcoma not otherwise specified”). There may also be skin metastases among these samples.

However, due to limited clinical documentation in GENIE there is no way for us to conclusively confirm this. It should here be noted that the mentioned Low *TERT*_p + UV group serves a very limited purpose in the study (it was referred to only in one sentence in Results, in reference to **Supplementary Fig. 2**, which we have now removed). This is because even in the High *TERT*_p cohorts, there are plenty of *TERT*-driver negative samples to be used as control (e.g. only about 2/3 of melanomas carry the -124 or -146 mutations), as presented in **Fig. 2b**.

In the end, given the fact that this group of samples adds little value to the study and that it is impossible to verify the correctness of SBS7/DBS1 in these samples (due to lack of more detailed sample subclassification in GENIE), we decided in the end to simply not define it at all in **Fig. 1**, instead grouping all the Low *TERT*_p samples together under one label.

We are glad this was brought up, as we feel it led to a useful overhaul of the SBS burdens analysis and presentation (**Fig. 1**). The Methods text has been updated accordingly.

3.3. Figure 3. These plots aim to show that the frequency distributions of driver and atypical hotspot mutations versus mutation burden are different. Can authors provide any statistics to support this claim (the text refers to correlation, but this is not formally tested)? Also, does the mutation burden shown here include total mutations in each group --this is, UV-light and non-UV

light caused mutations--? Considering that the authors' hypothesis is that atypical mutations are caused by UV-light, it would be relevant to replicate this analysis using SBS7/DBS1 mutation burden alone.

We did previously consider all mutations, as SBS signature deconvolution is not possible on the individual samples due to low mutations counts in GENIE capture panel data. However, this is a relevant point, and the analysis has now been improved in two ways to address this comment:

- Recurrent mutations (≥ 4) have been removed in calculating the burdens, to avoid bias from common drivers (e.g. BRAF, or the TERTp drivers themselves; previously only the latter were removed)
- Instead of total TMB, we now specifically consider the diPy C>T burden, i.e. the load of UV-compatible mutations.

Fig. 3 has been updated to reflect this, with new bin boundaries to accommodate somewhat lower counts after applying these filters. The overall conclusions remain unchanged.

Additionally, to address the request for added statistics, we have calculated correlation coefficients and corresponding *P*-values, which are now presented in **Fig. 3** as well:

3.4. Figure 5b. The results could be improved if the authors quantified and compared the differences in CPD signals across the experimental groups.

As suggested, we have now quantified the bands and have presented the quantitative data in **Supplementary Fig. 5**.

3.5. The methods describing the signature fitting need to provide additional details, such as the reference set of signatures used.

We have updated the Methods text to describe this in more detail, obviously reflecting the changes introduced in response to comment 3.3 above (row 316):

“Single base substitution (SBS) and double base substitution (DBS) mutation signatures were calculated using the “DeconstructSigs” R package (version 1.9.0 together with COSMIC SBS signatures version 2 and DBS signatures version 3) using a maximum of 5 signatures and with the “exome2genome” option, with remaining parameters set to their default values. Recurrent driver mutations (SNVs present in ≥ 4 samples and DNVs present in more ≥ 2 samples in a given cohort) were removed before the analysis.”

3.6. While the data to reproduce the computational analysis of GENIE samples is provided, the code does not seem to be available for this or other computational analyses.

The code underlying all figures and analyses has now been included as a zip archive. Note that the actual patient-level GENIE data cannot be redistributed and needs to be downloaded from Synapse before running the script. For this reason, Supplementary Data 1 has been removed from this submission.

Minor

3.7. As an additional control, it would be interesting to show if there are differences in the frequencies of TERTp atypical mutations in high UV-light exposed and low/no UV-light-exposed skin melanomas.

This may not be immediately obvious, but this is in fact presented in **Fig. 3**. The analysis shows only skin cancers but binned by the degree of UV mutagenesis (diPy C>T burden). As can be seen in this figure, the atypical mutations are lacking in low-UV skin cancers as expected while being highly frequent (reaching above 15% of samples) in high-UV samples. To make sure this message is not lost on the reader, we have now clarified this in Results (row 160):

“Even within the group of UV-associated skin cancers, the atypical mutations thus arise preferably in highly UV-mutated samples.”

3.8. In the second section of results, on page 5 line 130, it is mentioned that atypical mutations at -101, -100 and -97 add up to 44 samples, but according to Supplementary Table 3 this should be 24 samples.

The correct number is indeed 44; however, this was not trivial to read out of the table the way it was previously structured. The fact that counts for the atypical mutations were split up by co-occurrence with multiple different driver events may have complicated decoding of the numbers in this table. **Supplementary Table 3** has now been thoroughly revised and checked and is now presented in what we believe is a clearer way, with driver co-occurrence presented as one aggregated number. Further separation by exact driver event is provided within parentheses (explained in the table legend):

TERT p native ETS atypical		
-97_SNV	4	4 (4,0,0)
-100_SNV	8	7 (6,1,0)
-101/-100_DNV	2	1 (0,1,0)
-101_SNV	30	29 (14,14,1)

3.9. Figure 1. Given the low frequency of atypical *TERT*p mutations compared to driver mutations, it is difficult to evaluate differences in the frequency of atypical mutations across the cohorts. Perhaps some changes in the colour codes, or plotting these mutations alone in a different panel, could help.

We are thankful for this suggestion and have now restructured **Fig. 1** as suggested, presenting the atypical mutations in a separate panel. Additionally, we have improved the UV signatures analysis and presentation thereof, such that the SBS7+DBS1 contribution to the total mutation count is presented together using stacked bars:

3.10. Figure 2 caption. *TERT* coordinates are written as chr8:1295188-1295268 where it should be chr5:1295188-1295268.

Thank you for pointing this out – this has now been corrected.

3.11. Figure 4b. Would it be possible to show the same scale on the y axis for the different sets of reads to better understand the frequencies between both alleles? For example, the y axis in Case 1 *TERT*p driver wild-type reads could range from 0 to 10, as it is shown on driver mutant reads.

In this figure, the length of the axis is used to indicate the total number of reads for the mutant and wildtype sets. E.g. in Case 1, this explains why the negative axis goes to -33. We failed to describe this in the legend, and have now added an explanatory sentence (row 628):

“The total number of driver mutant and wild-type reads are indicated by the height of the positive and negative axes, respectively.”

3.12. Figure 6b is missing y axis labels.

We are grateful for the careful checking of the figures – this error has now been corrected.

3.13. Supplementary Table 1. There are different hg19 mutation coordinates that have been incorrectly mapped to hg38, including chr5:1295228.

The coordinates have been corrected – thank you for pointing this out.

3.14. Supplementary Table 3. The values of this table need to be reviewed and corrected. For example, counts and percentages shown for individual driver mutations do not add up to “All drivers” in any columns except in “Low TERTp no UV”. Likewise, the values within “All native and -146” seem to be incorrect based on the rows above.

This table has been restructured for improved readability – please see comment 3.8 above.